# Landscape Dynamics and Ecological Risk Assessment of Cold Temperate Forest Moose Habitat in the Great Khingan Mountains, China

**DOI:** 10.3390/biology12081122

**Published:** 2023-08-11

**Authors:** Shiquan Sun, Yang Hong, Jinhao Guo, Ning Zhang, Minghai Zhang

**Affiliations:** 1College of Wildlife and Protected Area, Northeast Forestry University, Harbin 150040, China; ssq2504204784@163.com (S.S.); hy1624@126.com (Y.H.); guojinhao19960206@126.com (J.G.); ningbala0327@163.com (N.Z.); 2Feline Research Center of National Forestry and Grassland Administration, Harbin 150040, China

**Keywords:** cold-temperate zone, moose, habitat, landscape pattern, landscape ecological risk

## Abstract

**Simple Summary:**

Through calculating landscape indexes and constructing an ecological risk assessment model, we assessed the landscape pattern and ecological risk of moose (*Alces alces*) habitat in the Nanwenghe National Nature Reserve of the Great Khingan Mountains in China. The results show that the areas with high habitat suitability for moose were mainly concentrated in forests and rivers. However, under the premise of global warming, the risk of landscape pattern fragmentation tends to increase from 2015 to 2020. Moose preference patch type is dispersed, the degree of polymerization is low, and the risk of patch type transformation is increased. In terms of ecological risk: the medium- and high-risk areas in the moose habitat are mainly concentrated in the river area and its nearby forests, showing a fine and scattered distribution. The low ecological risk area is mainly distributed on both sides of the road and the mountainous area, and the patch type is single. It shows that the moose habitat will have a weak ability to resist risk in the future. The study suggests that we should avoid further human intervention in forests and rivers, formulate reasonable forest protection and sustainable development plans in cold temperate zones, and effectively monitor and protect the dynamics of cold temperate forests and moose populations.

**Abstract:**

The change in habitat pattern is one of the key factors affecting the survival of the moose population. The study of the habitat landscape pattern is the key to protecting the Chinese cold-temperate forest moose population and monitoring the global distribution of moose. Through the ecological risk assessment of the moose habitat landscape pattern in a cold-temperate forest, we hope to assess the strength of habitat resistance under stress factors. This study provides a theoretical basis for the protection of the moose population in the cold-temperate forest in China and the establishment of the cold-temperate forest national park. In the study, the MaxEnt model, landscape index calculation and ecological risk assessment model construction were used to analyze the field survey and infrared camera monitoring data from April 2014 to January 2023. The habitat suitability layer of the moose population in the Nanwenghe National Nature Reserve of the Great Khingan Mountains was calculated, and the range of the moose habitat was divided based on the logical threshold of the model. The landscape pattern index of the moose habitat was calculated by Fragstats software and a landscape ecological risk assessment model was established to analyze the landscape pattern and ecological risk dynamic changes of the moose habitat in 2015 and 2020. The results showed that under the premise of global warming, the habitat landscape contagion index decreased by 4.53 and the split index increased by 4.86 from 2015 to 2020. In terms of ecological risk: the area of low ecological risk areas increased by 0.88%; the area of medium ecological risk areas decreased by 1.11%; and the area of high ecological risk areas increased by 0.23%. The fragmentation risk of the landscape pattern of the moose habitat tends to increase, the preferred patch type is dispersed, the degree of aggregation is low, and the risk of patch type transformation increases. The middle and high ecological risk areas are mainly concentrated in the river area and its nearby forests, showing a fine and scattered distribution. Under the interference of global warming and human activities, the fragmentation trend of the moose habitat in the study area is increasing, and the habitat quality is declining, which is likely to cause moose population migration. For this reason, the author believes that the whole cold temperate forest is likely to face the risk of increasing the transformation trend of dominant patch types in the cold-temperate coniferous forest region mainly caused by global warming, resulting in an increase in the risk of habitat fragmentation. While the distribution range of moose is reduced, it has a significant impact on the diversity and ecological integrity of the whole cold-temperate forest ecosystem. This study provides theoretical references for further research on the impact of climate warming on global species distribution and related studies. It is also helpful for humans to strengthen their protection awareness of forest and river areas and formulate reasonable protection and sustainable development planning of cold-temperate forests. Finally, it provides theoretical references for effective monitoring and protection of cold-temperate forests and moose population dynamics.

## 1. Introduction

A forest ecosystem is a system with a certain structure, function and self-regulation ability formed by the forest biological community and its environment. In the terrestrial ecosystem, the forest ecosystem has the largest biological community, the largest area, the most complex structure, and the most perfect function, which plays an indispensable role in the global ecological process [1]. As an important part of the terrestrial ecosystem, the forest ecosystem provides wild animals with food, habitat and other resources to ensure the survival and reproduction of wild animals [2,3]. Therefore, the stability of the function and structure of the forest ecosystem is important for the future of global biodiversity. Once the structure and function of the forest ecosystem is affected, it is likely to change the landscape pattern of wildlife habitats in the system. As the basis of species survival and reproduction, the change in landscape pattern can affect life history traits and survival of the species [4].

In recent years, under the disturbance of human activities to the global climate and forest ecosystem, the change in the species habitat landscape pattern caused by forest ecosystem degradation has affected the survival of species and become a global problem for wild animals [5,6,7]. For large forest mammals, habitat fragmentation is more likely to affect their access to food resources and safe places, and lead to an increase in individual mortality [8]. Based on the practice of wildlife conservation in recent years, it is particularly important to consider the dynamic changes of the ecosystem and habitat while carrying out the protection of target species. Therefore, it is of great significance for the protection of target species and forest ecosystems to understand the dynamic changes of the landscape spatial pattern of habitats and the ability of habitats to resist the disturbance of external factors.

The analysis of the landscape pattern of species habitats is based on the analysis of landscape patterns, combined with the preference of target species for landscape elements such as patch type, size and shape, and the results can effectively reflect the quality of the habitat, and to a certain extent, reflect the degree of species preference for the current habitat [9,10]. The ability of a habitat to resist external factors is based on the size of the landscape ecological risk index. The index indicates the effect and harm degree of negative factors such as an increased risk of habitat destruction and a reduction in biodiversity on regional ecosystem security caused by the dynamic change in the landscape pattern [11]. Combined with the preference of the target species for habitat selection, to a certain extent, the index not only reflects the habitat quality of target species, but also shows the harm to the forest ecosystem under a variety of stress factors such as human disturbance and natural disasters.

Located in the northeast of China, the Great Khingan Mountains is not only the main distribution area of the cold-temperate zone in China, but also an important sensitive area of global ecological change. It has the largest cold-temperate forest ecosystem in China, and its unique cold and humid environment has formed a large bright coniferous forest area with *Larix gmelinii* as the dominant tree species under the interaction of the permafrost, wetland and forest [12]. The Great Khingan Mountains forest ecosystem not only improves the global climate, but also provides conditions for many precious endangered species to survive and reproduce. Recent studies have shown that the cold-temperate zone is one of the main distribution areas of the permafrost on earth. Global warming will weaken or even lose the function of the permafrost, resulting in the release of stored soil organic carbon. While further aggravating greenhouse effect [13], it will lead to the increase in river flow level and water area in the cold-temperate forest, increase the frequency of flood disasters, and then change the species habitat pattern and affect the survival of species [14].

As one of the representative species of the cold-temperate forest ecosystem, moose population depends on the forest ecosystem to obtain survival resources. With the increasing risk of functional degradation of the forest ecosystem in the cold-temperate zone in recent years, it has produced a very significant ecological response to the size and distribution of the moose population. Moose distributed in the Great Khingan Mountains of China is the southernmost population of this species in Eurasia. Compared with moose populations in other distribution areas, the risk faced by the moose population in China is more severe [15]. On the one hand, due to global climate change, the negative impact of temperature change on moose is more intense at the southern edge of the species distribution, especially in moose, which are sensitive to temperature change, resulting in higher individual mortality [16]. Due to the enhanced disturbance of human activities in the forest ecosystem, the habitat area of moose decreased, the risk of fragmentation increased [17], and finally, led to moose population migration and even led to the decline in the population [18]. The local extinction of the population would not only reduce the distribution range of moose in China, but also affect the global distribution pattern of moose. Therefore, the landscape pattern dynamics of the cold-temperate forest moose habitat was analyzed, and the ecological risk assessment of the cold-temperate forest landscape is combined to evaluate the resistance of the habitat under stress factors. It can provide a theoretical basis for the protection of the cold-temperate forest moose population and the establishment of a cold-temperate forest national park in China.

To explore the potential risks of moose population and forest ecosystem in the cold-temperate zone under climate warming and human disturbance in the Nanweng River National Nature Reserve in the east of the Great Khingan Mountains, the land cover data of 2015 and 2020 were selected, combined with a field survey and infrared camera monitoring data from April 2014 to January 2023 to quantitatively evaluate the suitability of moose habitat, habitat landscape dynamics and ecological risk. The purpose of this study is to provide a theoretical basis for the efficient protection of species and habitat in cold-temperate forests.

## 2. Materials and Methods

### 2.1. Study Area

The study area chooses the Great Khingan cold-temperate Nanweng River National Nature Reserve, which is located in Songling District of the Great Khingan area of Heilongjiang Province, which is a branch of the Great Khingan Mountains, bounded by the Yilhuli Mountains in the north, Huma 12 stations in the east, and adjacent to the Jiagdachi Forestry Bureau in the south. The geographical coordinates are 51°05: 07″ N–51°39: 24″ E, 125°07: 55″–125°50: 05″ E, with a total area of 229,523 hm^2^, including forest area 128,137 hm^2^ and wetland area 70,480 hm^2^ (Figure 1). The land is a low mountain and hilly landform with little relief, high in the north and low in the south, high in the west and low in the east, with elevations of 500 m, 800 m, the lowest elevation of 370 m and the highest elevation of 1044 m (Figure 1). The reserve is located in the cold-temperate zone, which belongs to the continental monsoon climate zone of the cold-temperate zone, with a long cold period in winter and a short hot period in summer, with an annual average of −3 °C, a maximum temperature of 30 °C and a minimum temperature of −48 °C. The average annual precipitation is 600 mm. There are more than 20 large and small rivers in the territory, such as Nan Weng River, Kengdu River and Nanyang River, which flow through the whole territory from northwest to southwest and flow into Nenjiang River, which is the main birthplace and water conservation place of Nenjiang River. The soil types mainly include brown coniferous forest soil, meadow soil, swamp soil and stony soil. Typical brown coniferous forest soil is distributed on the middle slope of each slope direction, with an altitude of 700 m to 1000 m. Its vegetation type is a typical cold-temperate coniferous forest composed of *Larix gmelinii* as constructive species, including *Betula platyphylla*-*Larix gmelinii* forest, *Rhododendron dauricum*-*Larix gmelinii* forest, grass swamp and larch swamp. There are more than 800 species of plants belonging to 61 families. The main tree species are *Larix gmelinii*, *Betula platyphylla* and *Populus davidiana*, and the main shrub species are *Rhododendron dauricum* and *Spiraea salicifolia*. There are 74 families and 309 species of wildlife resources. Among them, herbivorous mammals mainly include moose, red deer (*Cervus canadensis*), Siberian roe deer (*Capreolus pygargus*) and *lepus timidus*. Carnivorous mammals mainly include *Lynx lynx*, *Gulo gulo* and so on. There are 216 species of birds, including *Tetrao urogalloides* and *Bonasa bonasia*. At the same time, there are 44 species of amphibians and reptiles such as *Salamandrella keyserlingii*.

### 2.2. Sample Line Data Acquisition

From December 2021 to January 2023, the sample line method was used in Nanwenghe National Nature Reserve. The length of the sample line was at least 4 km, and a large sample square of 10 m × 10 m was set up every 400 m and when the vegetation type changes, and five small sample squares of 1 m × 1 m were set in the sample square [19]. The survey area, date and time, sample coordinates, vegetation types, moose number and activity traces (footprints, feces and entities, etc.) were recorded. Sample line spacing was ˃2 km and sample lines were randomly arranged at different levels. Due to the lack of access by vehicles in some areas of the reserve, the sample lines in this area could not be arranged. In this survey, there were 126 sample lines, the number of samples was 1377, and the total length of sample lines was 512.6 km (Appendix A).

### 2.3. Infrared Camera Data Acquisition

From April 2014 to January 2023, infrared camera technology was used to monitor moose resources in Nanwenghe National Nature Reserve. Using the kilometer grid method, according to the 1 km * 1 km grid, the infrared camera was set up in the center of the grid where the moose may pass. In order to ensure the capture rate of the camera on the animals, the camera was bound to the trunk of the primary animal path when the camera was deployed. The height was 100–130 cm from the ground. Camera data was collected 4 times a year, and each time some camera positions were adjusted as needed. Camera shooting parameters were set to 2 consecutive shots, 5 s shooting interval and medium sensitivity. A total of 267 infrared cameras were set up in this survey [20]. According to the monitoring records, the infrared camera sites of moose entities were sorted out and summarized.

### 2.4. Environmental Data Acquisition

The distribution of moose is affected by climate, topography, human disturbance and other factors. In this experiment, 23 environmental factors were selected as modeling data, including landscape, topography, climate and human factors. They were 17 bioclimatic factors (Bio3–19), 2 geographical factors (elevation and slope), 2 vegetation data (landuse type and normalized difference vegetation index), 1 human disturbance factor (distance to road) and 1 basic geographic information data (distance to river). These data are listed in Appendix B. The spatial resolution of all environmental factors was resampled to 30 m, the study area was taken as the layer boundary of environmental variables, and the projection coordinate system 1 was set to WGS-1984-UTM-ZONE-50 N. It is convenient for the screening of subsequent environment variables.

### 2.5. Species Distribution Models Building

Through the collection of moose distribution point data of sample line survey records and infrared monitoring records, the data are preliminarily screened to eliminate data format errors and duplicate recorded data. After extraction and screening, the detailed longitude and latitude information of moose occurrence points were extracted, and a total of 145 moose occurrence sites were obtained. Using the SDMtoolbox spatial filtering tool in ArcGIS10.3, 1 km was selected as the sparse radius to filter the occurrence points [21], and 44 moose distribution points were obtained. In order to avoid the influence of the correlation between environmental variables on the results of the model, the Pearson correlation test of 23 environmental factors was carried out. Finally, environmental factors with a correlation coefficient of less than 0.7 were retained. After screening, 12 environmental factors were obtained (Table 1), and the obtained environmental factors were converted into ASC file format by using the transformation tool of ArcGIS10.3 [22]. The ROC curve (receiver operating characteristic curve), which is the value of the area under the subject’s working characteristic curve (AUC, area under curve), measures the pros and cons of the model. The screened environmental factors and field species survey data were imported into the Java’s MaxEnt model [23]. A total of 75% of the moose distribution sites were randomly selected as the training set to establish the prediction model. The rest of the distribution points were used as test sets to verify the model, and logistic was selected to output the results. The repeated operation coefficient of the model was set to 10 times, the repeated operation category selected sub samp, and other parameters used the default settings and ran [24].

### 2.6. Analysis of Landscape Pattern of Habitat

Based on the moose habitat simulated by MaxEnt model, the ArcGIS10.3 screening and extraction tools were used to cut the data of land cover types in 2015 and 2020. According to the attribute classification, the land cover types of moose habitat were divided into five first level types: arable land, forest, water area, grassland and unused land. There were 7 s-level types of dry land, closed forest land, shrubs, sparse woodland, high coverage grassland, medium coverage grassland and swamp [25,26]. Using Fragstats4.3 software, the following indicators were selected at the patch and landscape level to calculate the moose habitat landscape pattern index under different time series.

According to the landscape pattern index of 6 patch types [27], 6 indexes were selected and listed in Appendix C.

### 2.7. Ecological Risk Assessment of Habitat Landscape

Based on Fragstats4.3 software, the landscape pattern index of different patch types were obtained, and the landscape vulnerability index, landscape interference index and landscape loss index were calculated to construct the landscape ecological risk assessment model [28]. Based on the method of grid sampling, the fishing net map was compiled by ArcGIS10.3, and the landscape ecological risk index (ERI) in the grid was calculated. The ecological risk map of habitat landscape was compiled in ArcGIS10.3 by Kriging interpolation method [29].

Ecological risk index calculation formula is:(1)$$ERI=\sum_{i=1}^{n}\left(\frac{A_{ki}}{A_{k}}\right)*R_{i}$$
where $A_{ki}$ represents the category i landscape area of *k* sampling area. $A_{ki}$ represents the area of the sampling area. $R_{i}$ represents the landscape loss index.

The landscape loss index indicates the degree of loss when the landscape is disturbed, and its calculation formula is:(2)$$R_{i}=E_{i}*V_{i}$$
where $E_{i}$ indicates the landscape disturbance index. $V_{i}$ indicates the landscape vulnerability index. The method of expert assignment was used to assign the vulnerability of six landscape types, which was 6 for unused land, 5 for water area, 4 for arable land, 3 for grassland, 2 for forest and 1 for urban and rural construction land. Then, the landscape vulnerability index was obtained by normalization.

The landscape disturbance index indicates the degree of landscape disturbance, and its calculation formula is:(3)$$E_{i}=aL_{i}+bD_{i}+cF_{i}$$
where $L_{i}$, $D_{i}$, $F_{i}$ represent landscape fragmentation index, landscape division index and landscape fractal dimension index, respectively. A, b and c represent the weights of different landscape indexes, respectively. Combined with the actual situation of the study area [25], the weights (a, b, c) of landscape fragmentation index, landscape division index and landscape fractal dimension index were assigned as 0.5, 0.3 and 0.2, respectively.

Landscape fragmentation index indicates landscape internal stability, and its calculation formula is:(4)Li=niAi
where $A_{i}$ represents category *i* landscape area. $n_{i}$ represents the number of patches in category i landscape.

The landscape division index indicates the degree of spatial separation of patches, and its calculation formula is:(5)Di=(niA)∗A/2Ai
where A represents the total area of all landscapes.

The landscape fractal dimensional index represents the shape complexity and spatial stability of landscape patches, and its formula is:(6)Fi=2ln⁡(qi4)ln⁡Ai
where $q_{i}$ represents the perimeter of landscape type i.

## 3. Results

### 3.1. Habitat Suitability of Moose

The occurrence sites and environmental variables of the processed moose were introduced into the species distribution model to obtain the suitability layer of the moose in the study area (Figure 2). The simulation results of MaxEnt model show that the average training set AUC value is 0.828 and the standard deviation is 0.074, which indicates that the simulation result of species distribution model is better, and the fitting degree of the model is relatively high. Based on the Equal Training Sensitivity Specificity Threshold automatically generated by MaxEnt, the predicted map of moose habitat suitability index was converted into a binary map of habitat and non-habitat based on the output results of the model in ArcGIS10.3 software. The automatically generated Equal Training Sensitivity Specificity Threshold value in the results of the MaxEnt model is 0.435, which is used as a threshold to generate the distribution range of moose population in Nanweng River Nature Reserve (Figure 3), which is used as the boundary for subsequent landscape pattern analysis.

### 3.2. Landscape Dynamics of Moose Habitat

#### 3.2.1. Patch Scale Characteristics

Combined with the results of the field investigation, distribution model and landscape index calculation, the large area of moose habitat is mainly concentrated in the river area of Nanweng River and Guandu River, *Larix gmelinii*-*Betula platyphylla* mixed forest and grassland swamp area. Among them, the areas with the highest suitability are mainly rivers and larch forests (Figure 2 and Figure 3) to ensure the needs of moose individuals for food, temperature and other survival factors.

The largest landscape types in moose habitat are high coverage grassland and closed forest land, followed by sparse woodland and dry land, and the smallest area of swamp, shrub and medium coverage grassland (Figure 4 and Table 2). From 2015 to 2020, the patch area of dry land, shrub and sparse woodland increased, and the area of sparse woodland increased by 1159.11 hm^2^. The medium coverage grassland disappeared completely, and the patch type area of high coverage grassland, closed forest land and swamp decreased. Among them, the area of high coverage grassland decreased by 1683.09 hm^2^, the LPI of dry land and sparse woodland increased, and other landscape types decreased. In terms of LSI, dry land, closed forest land and sparse woodland increased by 1.16, 0.85 and 0.75,while shrubs, high coverage grasslands, medium coverage grasslands and swamps decreased by 0.42, 1.98, 1.4 and 2.12, respectively. In terms of FRAC_MN, shrubs and medium coverage grasslands decreased by 0.02 and 1.08, swamps increased by 0.03, and other landscape types changed little. The COHESION of shrub, dry land and sparse woodland increased by 0.78, 1.42 and 0.17, respectively, while that of closed forest land, high coverage grassland, medium coverage grassland and swamp decreased by 0.30, 0.22, 59.26 and 2.75, respectively.

#### 3.2.2. Landscape Scale Characteristics

From 2015 to 2020, the CONTAG decreased by 4.53, the SPLIT increased by 4.86, and the landscape fragmentation showed an increasing trend (Table 3). In the past five years, the AI changed little, and the SHDI and SHEI showed an increasing trend. The moose habitat is mainly in the vicinity of woodland, the Nanweng River and other rivers. From 2015 to 2020, the dominant patch area decreased, the landscape pattern fragmentation increased, and the closed forest land dominated by trees gradually changed into sparse woodland, dry land and other land types. Based on this, it shows that the whole cold-temperate coniferous forest region with coniferous forest as the zonal vegetation is likely to be in a similar transformation process. The change in patch type is likely to be part of the result and reflection of the northward movement of the whole cold-temperate zone under the condition of global warming.

### 3.3. Ecological Risk of Habitat Landscape

According to the natural breakpoint method, the landscape ecological risk area of moose habitat in the Nanweng River Nature Reserve is divided into three grades: low ERI (0.0196 ≤ ERI < 0.0248), medium ERI (0.0248 ≤ ERI < 0.0323), and high ERI (0.0323 ≤ ERI < 0.0587).

The results showed that the moose habitat in the Nanweng River Nature Reserve was dominated by low-risk area and stroke-risk area, the marginal woodland within 1 km on both sides of the highway belonged to low-risk area, and the ecological risk grade of high-coverage grassland and river area was higher. From 2015 to 2020, the proportion of areas with low ecological risk increased by 0.88%; that of areas with medium ecological risk decreased by 1.11%; and that of areas with high ecological risk increased by 0.23%. Combined with the moose habitat suitability index map, the areas with high suitability are mainly middle and high ecological risk areas. With the decrease in the distance from roads and building facilities, the ecological risk grade decreases and the suitability index decreases. The middle- and high-risk areas are concentrated in wetlands and water distribution areas, and the patches of closed forest land, swamp and high-coverage grassland are fine and scattered, and the characteristics of patch fragmentation are obvious. The low ecological risk area is mainly distributed on both sides of the road and in the mountainous area. Due to human activities, the land around and on both sides of the road is reclaimed into sparse woodland and dry land with a low canopy density, the patch type is single, and the level of ecological risk is low, and there is a tendency to expand outward on both sides of the highway in the low ecological risk area. Between 2015 and 2020, the area of high ecological risk areas increased, mainly due to the rise in river water levels in cold-temperate regions caused by global warming, resulting in an increase in regional risk index (Table 4 and Figure 5).

## 4. Discussion

As the basis for the survival and reproduction of the wild animal population, a habitat needs to meet its necessary survival conditions, including rich food resources, resting and breeding places, predation risk avoidance and other factors. Moose is species with strong adaptability to the cold and that is extremely sensitive to high temperatures [30]. In habitat selection, the habitat should not only meet the needs of its food resources and avoid natural enemies, but also ensure that the temperature conditions in the habitat meet the needs of species survival. Studies have shown that due to the high sensitivity of individual moose to temperature, when the temperature is higher than the critical threshold of the individual, the heart rate and metabolic rate of moose will increase exponentially. The physiological stress will be extremely great. This results in increased energy consumption, reduced food intake, and weight loss for individual moose. Eventually, mortality increases and even extinction occurs [31,32]. In addition, the effect of high temperature on the individual moose has a delay. In addition, the high and low temperature value and duration of the day have an impact on individual physiology. High temperatures from the previous day can limit a moose’s ability to lower its body temperature at night. When moose populations face high temperatures, in addition to adopting behavioral regulation strategies, the selection of a habitat that can relieve heat stress for a long time is particularly important for population survival [33,34]. On the other hand, annual changes in habitat nutrient conditions have a significant impact on moose population numbers. Oehlers, Seaton and others believe that good nutrition and high-quality food sources will meet the energy and protein needs of female moose while feeding their young, which in turn affects the birth rate of the young [35,36]. Many studies have shown that the average size and number of offspring are related to habitat nutritional conditions. In populations with good nutrition, females often choose to increase the number of young. When conditions are harsh, an increase in the average size of the offspring is chosen instead of an increase in number. Therefore, for moose species, habitat stability is not only related to the survival and distribution of the moose population, but also has a direct effect on population numbers [37,38]. Once the habitat pattern changes, it will directly affect the distribution of the moose population in the habitat [18,39].

The habitat pattern dynamics were analyzed to investigate its influence on the moose population, so as to implement ecological restoration and associated management measures in time. Based on the dynamic change in the landscape pattern, the potential risk of the current cold-temperate forest ecosystem under the influence of universal climate change and human activities were analyzed. In this paper, the occurrence point of moose was obtained by field investigation and infrared monitoring, the habitat range of moose was determined by the species distribution model, the pattern index of the patch level and landscape level was computed by Fragstats4.3 software, and the dynamic changes of the landscape pattern of moose habitat under different scales and different time series were analyzed. Landscape index reflects the spatial structure of different types of patches combined with species preference for patch types and landscape pattern changes, and to a certain extent, it can be used as one of the bases for judging the current habitat preference of animal populations.

According to the calculation results of the landscape index, from 2015 to 2020, on the patch scale, the patch area of moose preferred land cover type decreased, the patch density decreased, the patch shape became more complex, and the agglomeration degree of the main patch types was weak and the trend of dispersion increased. In terms of landscape scale, the degree of landscape aggregation of moose habitat in the reserve is high, and the degree of landscape separation is low, indicating that the degree of landscape fragmentation is not high on the landscape scale, but the CONTAG decreases and the SPLIT increases. It shows that the landscape fragmentation of moose habitat has an increasing trend.

In recent years, due to the rise in sea level and the increase in land water area caused by global warming, the water level of the Nenjiang River and its tributaries has increased higher than the warning line many times. An on-the-spot investigation found that serious floods occurred in the study area in 2016 and 2020. This shows that the dynamic change in the landscape pattern from 2015 to 2020 was mainly caused by frequent floods caused by global warming. Previous studies have shown that the increase in temperature affects the function and structure of the ecosystem, which in turn affects the distribution pattern of animals and plants [40]. Based on the results of landscape index, it shows that under the influence of global climate change, the risk of landscape pattern fragmentation in the Great Khingan forest region in the cold-temperate zone increases, and the dominant species, which are dominated by coniferous forest, begin to decrease, and the dominant type patches tend to disperse on the landscape scale. The area of woodland decreased, while the area of sparse woodland and dry land increased. Under the condition of global warming, the change in patch types and the decrease in dominant tree species are likely to be part of the result and embodiment of the northward movement of the cold-temperate zone. For the cold-temperate moose population, the negative factors such as the change in habitat pattern and the increase in cold-temperate temperatures will greatly affect the survival and reproduction of moose individuals [15,41].

According to the patch type change and landscape index calculation results, the moose habitat near the highway and other human buildings, the patch type is single, there is a trend of patch-type transformation, the original woodland and high coverage grassland changed into dry land and sparse woodland and spread to both sides along the highway. It shows that the direct interference of human activities in the natural environment is enough to aggravate the transformation of patch types in the habitat. Based on this, it shows that with global warming, direct human interference, natural disasters and other factors are likely to directly affect the stability of the whole cold-temperate forest ecosystem, and then affect the living environment of forest wildlife species such as moose in the cold-temperate zone.

In the face of external interference, the resistance and resilience of different types of patches are different. When the intensity and frequency of external interference increase, the risk of damage to some types of patches will increase, and even lead to a change in patch type, and then, affect the habitat landscape pattern, and finally, lead to the distribution of moose population in the habitat. Landscape ecological risk assessment focuses on identifying the hazards that may be caused by natural disasters and human activities in the assessment region to the study of regional receptors (land use, landscape pattern, etc.) [11,42]. An ecological risk assessment model is constructed, and the results can explain the changes of the spatial distribution pattern of risk under different time series and analyze the resistance of the habitat to external disturbance under the current risk distribution [43,44]. This paper takes the moose habitat as the modeling area, human activities such as road construction and natural disasters from 2015 to 2020 as the risk source, and the habitat landscape pattern as the risk receptor to construct the ecological risk assessment model of the cold-temperate forest moose habitat in the Great Khingan Mountains. From the results (Figure 5) of the temporal and spatial distribution of ecological risk intensity in 2015 and 2020, the increasing trend of risk intensity in moose habitat is not obvious; in terms of spatial evolution, the middle- and high-risk areas of the habitat are concentrated in areas far away from human activities, and the risk grade decreases gradually with the decrease in distance from the road. It shows that the ecological risk level of moose preference patch type is higher, and the ability to resist risk sources is weak compared with other types of patches. Some studies have shown that the areas with medium and high ecological risks are mainly concentrated in the distribution areas of rivers and wetlands [45].

According to the results of this risk model, the increase in the area of high-risk areas in the study area may be due to the expansion of river areas caused by floods, and the increase in risk value caused by the transformation of forests and grasslands into wetlands. Therefore, under the condition of global climate change, the risk value of the whole cold-temperate forest region, especially in the vicinity of the river, is likely to increase due to the change in surrounding patch types caused by the increase in the river area. The consequences usually reduce the ability of the region to resist the source of risk, which will further aggravate the patch-type transformation rate under the condition of external factors. At the same time, under the premise of global warming, the release of organic carbon from the permafrost in the cold-temperate zone amplifies the positive feedback effect on climate warming [46,47]. And, the two factors promote each other to aggravate the risk value in some areas of the cold-temperate zone, which leads to the weakening of the resistance to external factors in the cold-temperate zone and finally affects the distribution of moose population in the cold-temperate zone.

## 5. Conclusions

This study shows that the whole cold-temperate region is facing the risk of increasing the transformation trend of dominant patch types in the cold-temperate coniferous forest region dominated by coniferous forest caused by global warming, and even the whole cold-temperate zone may move northward. This not only leads to the increase in population mobility and mortality due to the decrease in habitat suitability and the increase in environmental temperature of moose populations in the south of cold-temperate zone (Figure 6). It is also likely to cause a lack of diversity and ecological integrity in the whole cold-temperate forest ecosystem [48,49]. In addition, the direct interference of human activities with the natural environment also aggravates the speed of the patch-type transformation in cold-temperate forest areas, especially in the vicinity of highways and building facilities. However, in this paper, human activity is low in the study area. Under global warming conditions, the type and rate of patch-type transformation may be different from that in other regions with higher human activity. For example, in farmland areas, human activities are likely to directly change the habitat of species. This makes the conclusion of this paper biased in different regions.

We suggest that strengthening the protection of forests and rivers and decreasing the intensity of human activities in protected areas can effectively reduce the disturbance and negative impact of human activities on moose resting and feeding places. We should formulate a reasonable plan for forest protection and sustainable development in the cold-temperate zone, and reasonably plan the area of wildlife habitat and eco-tourism scenic spots, slow down the process of patch transformation, fully restore the connectivity of moose habitat, and better protect the cold temperate forest moose habitat.

## Figures and Tables

**Figure 1 biology-12-01122-f001:**
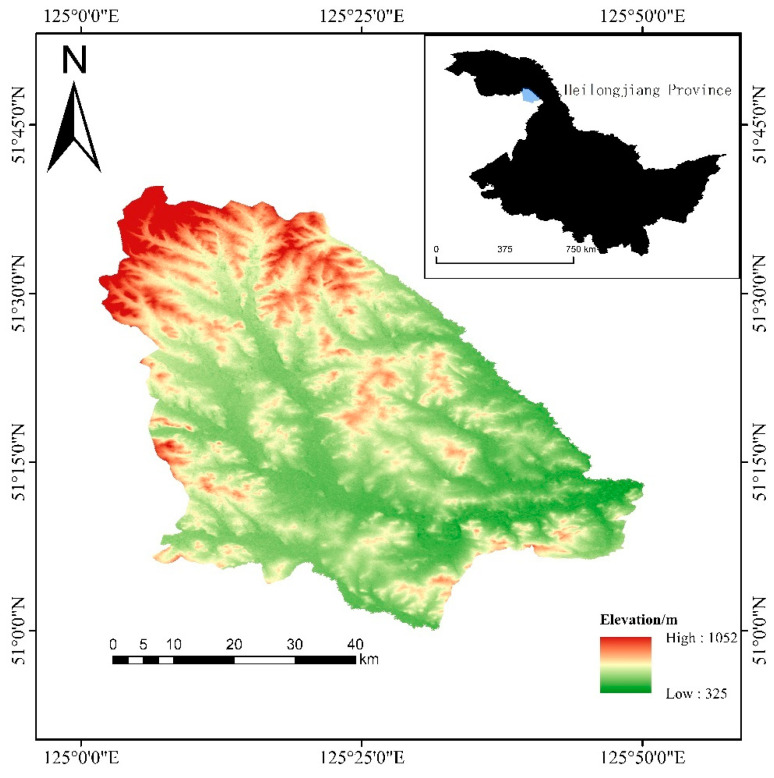
The map of Nanweng River national wetland protection area.

**Figure 2 biology-12-01122-f002:**
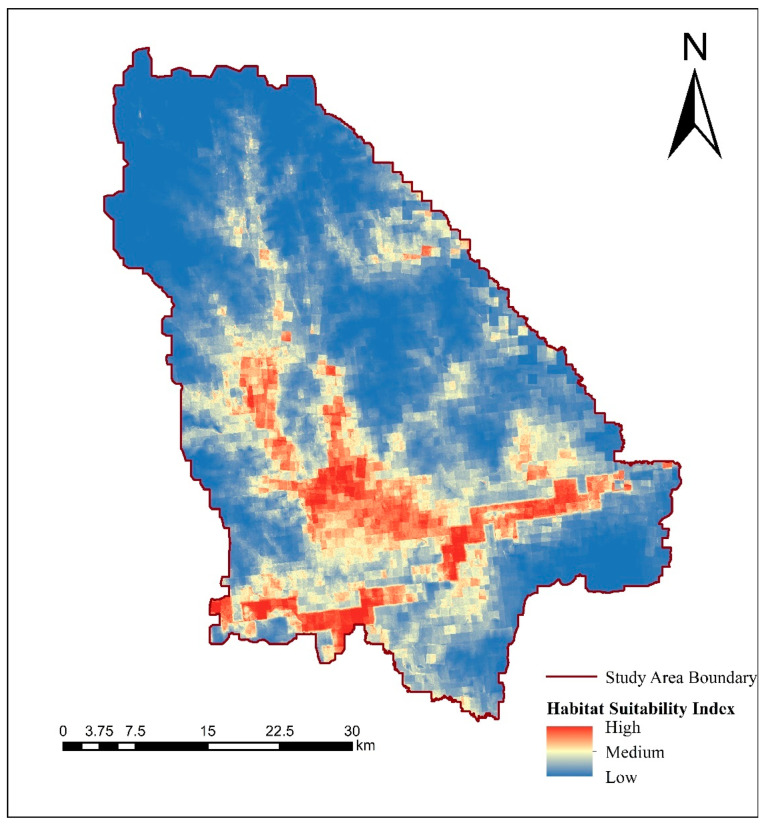
Habitat suitability index of moose in the study area.

**Figure 3 biology-12-01122-f003:**
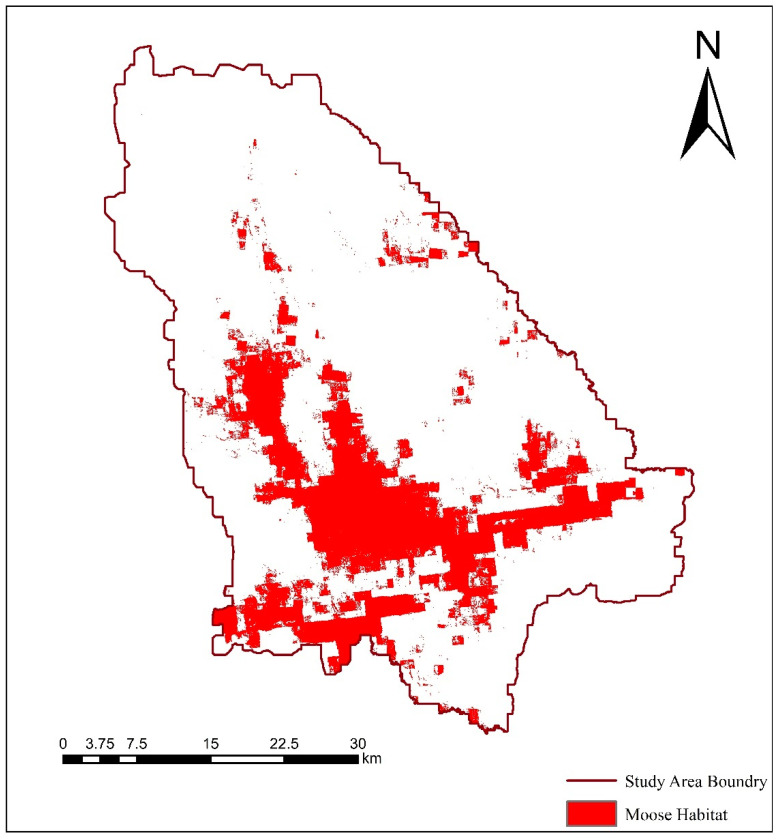
Habitat patches of moose in the study area.

**Figure 4 biology-12-01122-f004:**
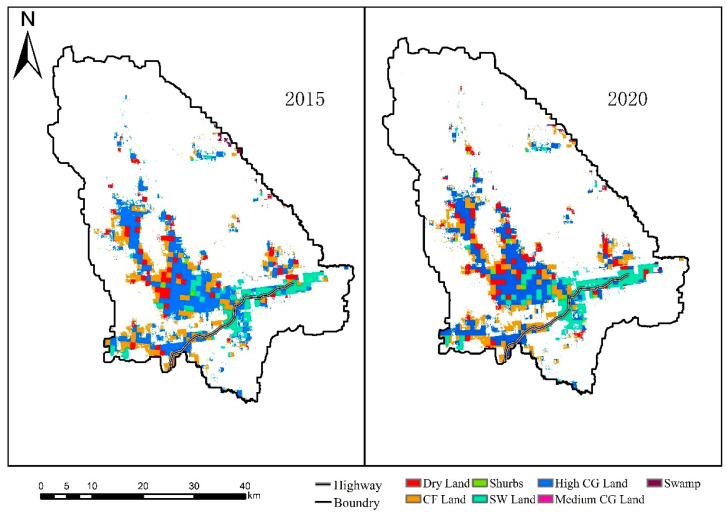
Landscape types of the study area.

**Figure 5 biology-12-01122-f005:**
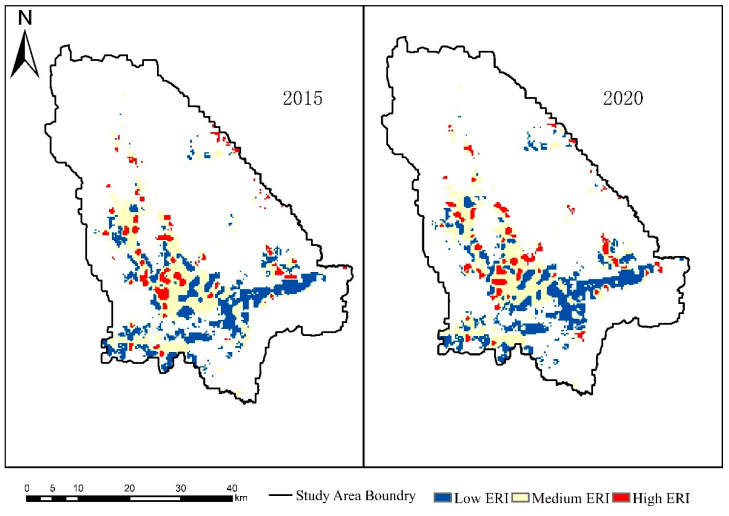
Spatial distribution of different degrees of landscape ecological risks in the study area during 2015–2020.

**Figure 6 biology-12-01122-f006:**
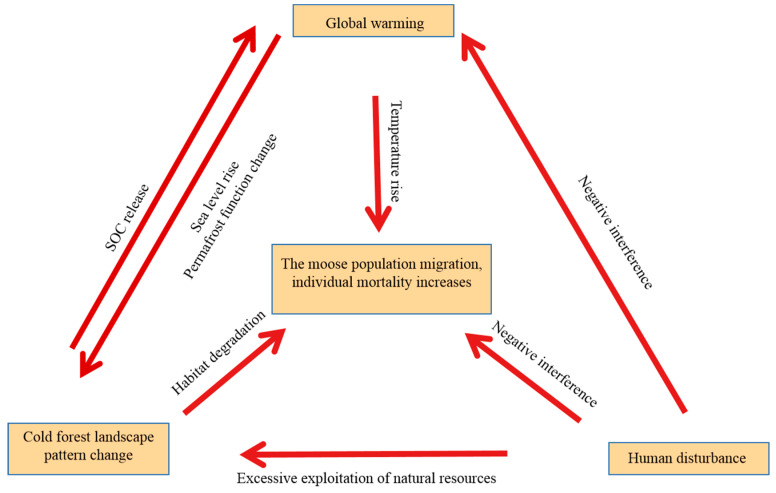
Potential threatening factors and their relationships of moose in cold-temperature forest.

**Table 1 biology-12-01122-t001:** Environmental variables used to model the range of moose.

Variable	Description	Data Source
Bio3	Isothermality/°C	WorldClim-Global Climate Data (https://chelsa-climate.org) (accessed on 30 October 2022)
Bio9	Mean daily mean air temperatures of the driest quarter/°C	WorldClim-Global Climate Data (https://chelsa-climate.org) (accessed on 30 October 2022)
Bio11	Mean daily mean air temperatures of the coldest quarter/°C	WorldClim-Global Climate Data (https://chelsa-climate.org) (accessed on 30 October 2022)
Bio14	Precipitation amount of the driest month/kg m^−2^	WorldClim-Global Climate Data (https://chelsa-climate.org) (accessed on 30 October 2022)
Bio15	Precipitation seasonality/kg m^−2^	WorldClim-Global Climate Data (https://chelsa-climate.org) (accessed on 30 October 2022)
Bio17	Mean monthly precipitation amount of the driest quarter/kg m^−2^	WorldClim-Global Climate Data (https://chelsa-climate.org) (accessed on 30 October 2022)
Dem	Elevation/m	https://www.gscloud.cn/ (accessed on 30 October 2022)
Sl	Slope/°	https://www.gscloud.cn/ (accessed on 30 October 2022)
Ndvi	Normalized difference vegetation index	https://www.gscloud.cn/ (accessed on 30 October 2022)
Highway	Distance to highway	https://www.webmap.cn/ (accessed on 30 October 2022)
Distw	Distance to water	https://www.webmap.cn/ (accessed on 30 October 2022)
LC	Land cover	https://www.resdc.cn/ (accessed on 30 October 2022)

**Table 2 biology-12-01122-t002:** Landscape patch features of the study area.

First Level of Land Use	Second Level of Land Use	Year	CA	PD	LPI	LSI	FRAC_MN	COHESION
Arable land	Dry land	2015	5196.6	0.8394	1.1106	17.8295	1.0526	96.3978
		2020	5759.19	0.8369	1.4646	18.9881	1.0516	97.1821
Forest	Closed forest land (CF Land)	2015	12,057.57	2.7739	3.1891	33.6194	1.0486	97.7904
		2020	11,833.2	2.7595	2.743	34.4656	1.0505	97.4829
	Shrubs	2015	327.78	0.0837	0.2219	5.2893	1.057	94.5468
		2020	618.66	0.0617	0.2218	4.8675	1.0389	95.9629
	Sparse woodland (SW Land)	2015	7268.22	0.8372	7.7344	15.3779	1.0498	98.8462
		2020	8427.33	0.8215	8.0917	16.1256	1.0476	99.0149
Grassland	High coverage grassland (HighCG Land)	2015	20,134.89	2.7056	19.6771	30.7685	1.05	99.2258
		2020	18,451.8	2.667	16.5345	28.7903	1.0514	99.0047
	Medium coverage grassland (MediumCG Land)	2015	0.54	0.0022	0.0012	1.4	1.0831	59.2586
		2020	0	0		0	0	0
Unused land	Swamp	2015	402.03	0.1432	0.4073	10.8806	1.0523	96.4141
		2020	316.17	0.0749	0.1304	8.7647	1.0836	93.6593

**Table 3 biology-12-01122-t003:** Landscape features of the study area.

Year	CONTAG	SPLIT	AI	SHDI	SHEI
2015	63.1966	18.0312	93.0191	1.3318	0.6844
2020	58.6675	22.8945	93.0996	1.384	0.7724

**Table 4 biology-12-01122-t004:** Proportion of different degrees of ecological risks in study area.

Year	Low ERI	Medium ERI	High ERI
2015	40.86	46.85	12.29
2020	41.74	45.74	12.52

## Data Availability

Our study did not create new data, and all data sources are available for free download from the attached table.

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
