# Peer review of "Landscape Dynamics and Ecological Risk Assessment of Cold Temperate Forest Moose Habitat in the Great Khingan Mountains, China"

_biology, 2023, doi:10.3390/biology12081122_

Round 1

Reviewer 1 Report

Line 13: Change ‘is’ to ‘was’

Line 28: Starting with ‘By means…’. Run-on sentence, reformat

Line 41: Remove ‘And’

Line 84: Capitalize the ‘T’ in ‘the’ at the start of the sentence.

Line 92: Merge the two paragraphs.

Line 114: Remove ‘On the other hand’, idiom used in prior sentence

Line 168: A figure showing the sample line layout would be helpful to visualize sampling design

Line 179: Photo acquisition rates? How frequently were photos taken?

Line 186: Any citations for the data sources?

Line 230: It is not necessary to include the equations for landscape metrics in text. If you want to include them, please do in an appendix at the end

Line 336: Says Fig2, should be 3?

Line 342: should be km2 not hm2?

Line 359: ‘From 15’, use ‘2015’

Line 398: Figure 5 caption needs more detail

Line 408: Run-on sentence

Line 431: Capitalize ‘on’

Line 528: References need standardized formatting. Some names are capitalized, others are not. Some have hanging indent, others do not

I think the Interspersion is a better measure of fragmentation than contagion in that at that small spatial scale (30 x 30) classification error can create misleading results. Additionally, moose are perceiving habitat at that fine of scale, they are using patches of forest, patches of openings, so interspersion would be more reflective of how moose use the landscape.

The discussion lacks depth. The authors reference changes in landscape structure and how that could potentially impact moose due to climate change, but the author never really addresses the functional biology of moose and how they directly interact with their environment. The authors don’t acknowledge behavioral plasticity in moose and don’t seem to grasp actual known habitat use. There is a such a deep pool of literature on moose resource use and very little of it is reference. A prime example is in the last sentence where the author suggests that some of the high risk areas along riparian zones could transition away from forests into wetlands due to flooding, but moose often utilize wetlands for thermoregulation and forage, and will likely become a more important habitat should the thermal stressors increase with climate change. As it stands, the paper is an assess of landscape change over time, and the target species seems irrelevant. A deeper dive into how these habitat changes could impact moose is needed.

Author Response

First of all thank you extremely greatly for your positive feedback and suggestions on this article.We have edited, corrected and improved the manuscript.And reply to the review experts' opinions one by one.

Line 13: Change ‘is’ to ‘was’

Thank you for the suggestion, we have changed the word to “to was”.

Line 28: Starting with ‘By means…’. Run-on sentence, reformat

Thank you for the suggestion, this sentence has been reformed.

Line 41: Remove ‘And’

Thank you for the suggestion,the word has been removed.

Line 84: Capitalize the ‘T’ in ‘the’ at the start of the sentence.

Thank you for the suggestion, this change has been made.

Line 92: Merge the two paragraphs.

Thank you for the suggestion, we have merged the two paragraphs.

Line 114: Remove ‘On the other hand’, idiom used in prior sentence

Thank you for the suggestion, the word has been removed.

Line 168: A figure showing the sample line layout would be helpful to visualize sampling design

Thank you for the suggestion! We have added a figure for the sample line. The figure are listed in figure S1.

Line 179: Photo acquisition rates? How frequently were photos taken?

Thank you very much for your advice.We have added information about camera parameters in line 188 of the text.The camera takes 2 consecutive shots, 5s shooting interval, medium sensitivity.

Line 186: Any citations for the data sources?

Thank you very much for your advice.Information about the sources of all environment variables has been tabulated and placed in the appendix.The appendix is on line 628.

Line 230: It is not necessary to include the equations for landscape metrics in text. If you want to include them, please do in an appendix at the end.

Thank you very much for your advice.At your suggestion, we have also considered that it is not necessary to present the equations in article.Therefore, we make a table and put it in the appendix.

Line 336: Says Fig2, should be 3?

Thank you for your advice.

In this part, Figure 2 is used as a map of the suitability index of moose in the study area.It is the source of Figure 3.We hope that the reader will read the two images together to help them understand the following content.

Line 342: should be km2 not hm2?

Thank you for your advice.Hectares are right.The article refers to the area data, and they are all in hectares.

Line 359: ‘From 15’, use ‘2015’

Thank you for the suggestion, this change has been made.

Line 398: Figure 5 caption needs more detail

Thank you very much for your advice.We didn't take into account the details of the caption.Now we have completed the modification.

Line 408: Run-on sentence

Thank you for the suggestion,we have reformed the sentence.

Line 431: Capitalize ‘on’

Thank you for the suggestion, this change has been made.

Line 528: References need standardized formatting. Some names are capitalized, others are not. Some have hanging indent, others do not

Thank you very much for your advice.We re-corrected the format of the reference.Now the format has been modified.

I think the Interspersion is a better measure of fragmentation than contagion in that at that small spatial scale (30 x 30) classification error can create misleading results. Additionally, moose are perceiving habitat at that fine of scale, they are using patches of forest, patches of openings, so interspersion would be more reflective of how moose use the landscape.

The discussion lacks depth. The authors reference changes in landscape structure and how that could potentially impact moose due to climate change, but the author never really addresses the functional biology of moose and how they directly interact with their environment. The authors don’t acknowledge behavioral plasticity in moose and don’t seem to grasp actual known habitat use. There is a such a deep pool of literature on moose resource use and very little of it is reference. A prime example is in the last sentence where the author suggests that some of the high risk areas along riparian zones could transition away from forests into wetlands due to flooding, but moose often utilize wetlands for thermoregulation and forage, and will likely become a more important habitat should the thermal stressors increase with climate change. As it stands, the paper is an assess of landscape change over time, and the target species seems irrelevant. A deeper dive into how these habitat changes could impact moose is needed.

First of all, thank you very much for your suggestions for this article. They are of great help in improving the paper.

In order to solve the problem of scale,we think that compared with the large spatial scale, the small spatial scale land use, NDVI and other data will be more accurate.In this way, in the subsequent construction of the species distribution model, the results will be more accurate.In addition, this paper analyzes the landscape pattern dynamics of moose by determining its habitat range.Therefore, the accuracy of model results is very important for full-text data analysis and discussion. Therefore, in order to ensure the accuracy of the model results, we use the data of small spatial scale for analysis and discussion.

On the other hand, we didn't really get into the moose interacting with the habitat in the first place.This includes fewer citations of relevant literature.This also makes our results less relevant to the moose species themselves.Therefore, we have adjusted and rewritten the discussion.In this article, we present the work of other scholars to help readers understand the impact of habitat change on moose species.We cited two aspects as fellows:

  • Habitat helps moose populations relieve heat stress.
  • Changes in nutrient conditions of habitat directly affect the number of moose population.

We hope to demonstrate the importance of habitat to moose populations.

To concretely show that habitat change can have a huge impact on moose.

Thanks again for your positive feedback on this article.

While helping us to revise the article, it also helps us to improve the rigorous awareness of academic and writing.

Reviewer 2 Report

The authors present an interesting study on Landscape dynamics and ecological risk assessment for the habitat of Alces alces in China. The bright side of the manuscript is to provide practical details for the related topic. However, some points are missing (mentioned below) in the manuscript and some parts of the manuscript are not easy to understand. Because of these reasons, major concerns are raised. Therefore, I would like to make some suggestions to improve the quality of the paper as below:

General Comments

Some parts of the manuscript are not easy to understand (mentioned below in specific comments). There are many long sentences and wordiness. This situation disrupts the flow of the subject and the continuity of the reading. Because of this reason, authors should re-reconsider writing some parts of the manuscript.

The Introduction section needs structural changes. The methods section should be cited with relevant references and also needs structural changes. References for each method and software used for the study should be added to the Materials and Methods section.

The Discussion section should be enriched with a more theoretical interpretation and relate the present results with additional concepts. For instance, the study results can be discussed in the framework of habitat loss and fragmentation, and climate change in different species from different countries in the broader context. In this context, the discussion section also needs structural changes.

Moreover, the limitations of the study should be given in the conclusion section.

Specific Comments

Summary

Lines 11-16: Please rephrase this sentence.

Abstract

I think the abstract needs to be rephrased and improved. In my opinion, it is good to start with the problem examined in the study. Within this context, the main problem that is examined by the authors should be explained in 1-2 sentences at the beginning of the abstract. After that, the methods and the main results should be given briefly. This can be followed by the main findings of the study. Finally, what is the importance of the results and how the results contribute to further studies should be written down. In my opinion, it is always good to finish the abstract with such a sentence. Furthermore, authors may also say in 1-2 sentences that their findings contribute to the conservation of species and its habitat. In this way, the bridge between the problem and the solution found by the authors would be stronger.

Line 30: the author -> we

Introduction

Lines 58-63: Needs references.

Line 65: “affects the global biodiversity” -> “important for the future of global biodiversity”.

Lines 67-68: “the change of landscape pattern will directly affect the migration, diffusion and communication of species” –> “the change of landscape pattern can life history traits and survival of the species.”

Lines 69-71: Need references.

Line 90-92: “Located in the northeast of China, the Great Khingan Mountains is not only the main distribution area of cold temperate zone in China, but also an important sensitive area of global ecological change” Please delete this sentence or merge with the paragraph below (lines 93-103).

Lines 124-129: Please rephrase here. This part of the paper is important since the authors should explain the purpose of the study and their hypothesis (I mean; what is the problem and what did you do to solve this problem) are given here. Please explain the purpose of your study with shorter sentences.

Materials and Methods

Please add the related references for each method (Infrared camera, environmental data etc..) and software used for the study.

Lines 131-165: “2.1 study area” This part of the paper is too long. Please rephrase and delete all unnecessary sentences.

Lines 179-185: 2.3. “Infrared camera data acquisition” Please provide more details for the method. I mean, how many cameras were placed? How many days the cameras were used? Did any procedure were used to place cameras? Brief information on the technical detail of cameras is also needed. Related references should also be added.

Conclusion

Lines 186-195: “2.4. Environmental data acquisition” Please explain how environmental data were gathered and used by referring the Table 1. Related references should also be added.

Line 508: “Through this study” please delete. “the author suggests” -> we suggest

Lines 508-510: Please rephrase here.

Some parts of the manuscript are not easy to understand . There are many long sentences, gramatical mistakes and wordiness. 

Author Response

First of all thank you extremely greatly for your positive feedback and suggestions on this article.We have edited, corrected and improved the manuscript.And reply to the review experts' opinions one by one.

The authors present an interesting study on Landscape dynamics and ecological risk assessment for the habitat of Alces alces in China. The bright side of the manuscript is to provide practical details for the related topic. However, some points are missing (mentioned below) in the manuscript and some parts of the manuscript are not easy to understand. Because of these reasons, major concerns are raised. Therefore, I would like to make some suggestions to improve the quality of the paper as below:

General Comments

Some parts of the manuscript are not easy to understand (mentioned below in specific comments). There are many long sentences and wordiness. This situation disrupts the flow of the subject and the continuity of the reading. Because of this reason, authors should re-reconsider writing some parts of the manuscript.

The Introduction section needs structural changes. The methods section should be cited with relevant references and also needs structural changes. References for each method and software used for the study should be added to the Materials and Methods section.

The Discussion section should be enriched with a more theoretical interpretation and relate the present results with additional concepts. For instance, the study results can be discussed in the framework of habitat loss and fragmentation, and climate change in different species from different countries in the broader context. In this context, the discussion section also needs structural changes.

Moreover, the limitations of the study should be given in the conclusion section.

Thank you for your positive feedback and suggestions on this manuscript. We have edited and polished the manuscript to improve the scientific language, and replied to the reviewers’ comments point-by-point.

Specific Comments

Summary

Lines 11-16: Please rephrase this sentence.

Thank you for the suggestion, we have rephrased this sentence.

Abstract

I think the abstract needs to be rephrased and improved. In my opinion, it is good to start with the problem examined in the study. Within this context, the main problem that is examined by the authors should be explained in 1-2 sentences at the beginning of the abstract. After that, the methods and the main results should be given briefly. This can be followed by the main findings of the study. Finally, what is the importance of the results and how the results contribute to further studies should be written down. In my opinion, it is always good to finish the abstract with such a sentence. Furthermore, authors may also say in 1-2 sentences that their findings contribute to the conservation of species and its habitat. In this way, the bridge between the problem and the solution found by the authors would be stronger.

Line 30: the author -> we

Thank you for the suggestion,We have revised and adjusted the entire abstract.

We express the main research questions at the beginning of the abstract to help readers better understand the purpose of the article.

In addition, further research and discussion of the results are also indicated in the abstract.

Thank you for the suggestion, we have changed the word to “we”.

Introduction

Lines 58-63: Needs references.

Thank you for your advice, we have added references.

Line 65: “affects the global biodiversity” -> “important for the future of global biodiversity”.

Thank you for the suggestion, we have changed the sentence to “important for the future of global biodiversity”.

Lines 67-68: “the change of landscape pattern will directly affect the migration, diffusion and communication of species” –> “the change of landscape pattern can life history traits and survival of the species.”

Thank you for the suggestion, this change has been made.

Lines 69-71: Need references.

Thank you for your suggestions to help us supplement the text and references.

Line 90-92: “Located in the northeast of China, the Great Khingan Mountains is not only the main distribution area of cold temperate zone in China, but also an important sensitive area of global ecological change” Please delete this sentence or merge with the paragraph below (lines 93-103).

Thank you for the suggestion, we have merged the sentence with the paragraph below(Line 96-104).

Lines 124-129: Please rephrase here. This part of the paper is important since the authors should explain the purpose of the study and their hypothesis (I mean; what is the problem and what did you do to solve this problem) are given here. Please explain the purpose of your study with shorter sentences.

Thank you for the suggestion,we have rephrased that. And we added the research question and purpose(Line 130-136).

Materials and Methods

Please add the related references for each method (Infrared camera, environmental data etc..) and software used for the study.

Thank you very much for your advice.We supplemented the literature on all the research methods used.To ensure that readers can obtain the specific operation of the method through literature.

Lines 131-165: “2.1 study area” This part of the paper is too long. Please rephrase and delete all unnecessary sentences.

Thank you for your suggestions. We have rephrase that part and deleted all unnecessary sentences.We hope to illustrate the main dominant tree species in the area by illustrating the flora and fauna of the area.And indirectly indicate the animal composition of moose in the study area.Helps readers identify potential threats to moose populations in the region (for example:the types of natural enemies)

Lines 179-185: 2.3. “Infrared camera data acquisition” Please provide more details for the method. I mean, how many cameras were placed? How many days the cameras were used? Did any procedure were used to place cameras? Brief information on the technical detail of cameras is also needed. Related references should also be added.

Thank you for your suggestions.We have added camera parameter Settings, as well as the frequency of photo data collection(Line 182-192).

Conclusion

Lines 186-195: “2.4. Environmental data acquisition” Please explain how environmental data were gathered and used by referring the Table 1. Related references should also be added.

Thank you for your suggestions.We have tabulated all relevant environmental variable data in appendix 1.

Line 508: “Through this study” please delete. “the author suggests” -> we suggest

 Thank you for the suggestion, we have changed the word to “we suggest”.

And we have deleted “Through this study”.

Lines 508-510: Please rephrase here.

Thank you for the suggestion, this change has been made.

Some parts of the manuscript are not easy to understand . There are many long sentences, gramatical mistakes and wordiness. 

Thank you very much for your positive feedback.We have corrected and modified the sentences in the article.Thank you again for your suggestions to help us improve the accuracy and language fluency of our articles.

Round 2

Reviewer 1 Report

The author has completed appropriate changes and improved the overall quality of the manuscript.

Reviewer 2 Report

The authors improved the manuscript with the previous comments.